# Perspective of the World Rehabilitation Alliance: Global Strategies to Strengthen Spinal Cord Injury Rehabilitation Services in Health Systems

**DOI:** 10.3390/healthcare12222313

**Published:** 2024-11-20

**Authors:** Colleen O’Connell, Jo Armstrong, Roger De la Cerna-Luna, Suvarna Ganvir, Paula Arnillas Brigneti

**Affiliations:** 1Stan Cassidy Centre for Rehabilitation, Dalhousie Medicine New Brunswick, Faculty of Medicine, Dalhousie University, Fredericton, NB E3B 0C7, Canada; 2National Spinal Injuries Centre, Stoke Mandeville Hospital, Bucks HP21 8AL, UK; armstrongjoc@gmail.com; 3Physical Medicine and Rehabilitation Department, Hospital Nacional Edgardo Rebagliati Martins, Lima 15072, Peru; rdelacerna89@gmail.com (R.D.l.C.-L.); paula.arnillas.b@upch.pe (P.A.B.); 4DVVPF’s College of Physiotherapy, Maharashtra University of Health Sciences, Ahmednagar 414111, Maharashtra, India; suvarna.ganvir@gmail.com; 5Primary Care Task Force, Indian Federation of Neurorehabilitation (IFNR), Mumbai 400020, Maharashtra, India

**Keywords:** spinal cord injuries, rehabilitation, sustainable development, emergencies, workforce, health services

## Abstract

Background/Objectives: Spinal cord injury (SCI) is a disabling condition prevalent worldwide, requiring rehabilitation services from injury through community living. This study, conducted by representatives of the World Rehabilitation Alliance (WRA), aims to identify strategies for strengthening SCI rehabilitation services globally, with particular attention to settings where resources are limited. Methods: Three focus groups were held between 2023 and 2024 with WRA representatives specializing in SCI rehabilitation. Discussions focused on four key areas: workforce and education, health policy and systems research, primary care, and emergency response. Perspectives were developed taking into account frameworks from the World Health Organization (WHO). Results: Key insights into SCI rehabilitation services emphasize workforce and education as critical areas, underscoring the importance of specialized training, certification, and ongoing support to build capacity. In health systems and policy research, significant gaps in evidence-based practices were highlighted, emphasizing the need for comprehensive data collection and national registries to guide policy and align SCI care with global standards. The integration into primary care systems is recommended to improve access and address common complications in low- and middle-income countries (LMICs). For emergency response, this study stresses the importance of preparedness and establishing multi-disciplinary teams capable of managing SCI cases in resource-limited settings, reducing preventable complications, and improving patient outcomes. Conclusions: SCI rehabilitation services are essential to global health, with a need for workforce development, research, national registries, and integration into primary and emergency care. Such efforts should improve accessibility and align with global best practices, ensuring comprehensive and accessible rehabilitation for all.

## 1. Background: Current State of SCI Rehabilitation

Spinal cord injury (SCI) is a devastating, life-changing event that can lead to significant disability, as well as socioeconomic challenges for the individual, their family, and community [1]. Prompt, appropriate intervention is critical to reducing the likelihood of avoidable secondary complications, particularly urinary tract infections, pressure injuries, joint contractures, and autonomic problems. Individuals with suspected or confirmed SCI should be admitted to a trauma center where they can be medically stabilized and assessed [2]. Following confirmation of a definitive cord injury, national guidance recommends referral to a designated SCI rehabilitation center according to the center’s agreed geographical catchment area, and registration with a national database for SCI [3]. However, not all individuals with SCI require inpatient rehabilitation; many with lower-level injuries, such as cauda equina syndrome (CES), will be supported through outpatient services and national charities, where these exist [4].

The associated immobility and lack of physical activity contribute to a long-term increased risk of fractures, as well as metabolic changes that may lead to the development of diabetes and cardiovascular disease [2,5]. The sensory impairment may be associated with the development of chronic neuropathic pain and can lead to skin breakdown (pressure injuries), which carry significant morbidity and mortality [2]. Autonomic impairments result in a reduced ability to regulate one’s blood pressure, body temperature, and breathing [2]. For this reason, ongoing access to medical services is needed for people with SCI to manage secondary complications if they arise, monitor their urological health, reassess their need for assistive devices, and support them in aging with SCI [4]. SCI rehabilitation is optimally delivered through a specialized multidisciplinary team, including but not limited to Physical Medicine and Rehabilitation (PM&R), nursing specialized in neurogenic bowel and bladder management, physiotherapy, occupational therapy, peer educators, speech and language therapy, and sports therapy, with access to associated specialties such as plastic surgery, urology, orthopedic surgery, and related investigations [5]. Without rehabilitation, substantive disability and mortality rates following SCI are very high [6], whereas following specialist rehabilitation, life expectancy in high-income countries can approach between 70 and 85% of the general population depending on the level of the injury [7].

The incidence of SCI varies from country to country, with a global range from 49 per million (New Zealand) to 8 per million (Spain); the prevalence ranges from 906 per million (USA) to 250 per million (France) [8]. The causes of SCI vary by country, with countries that have experienced natural disasters or conflicts often having a higher incidence, especially following earthquakes or prolonged conflicts affecting civilian populations [9]. The Nepal, Haiti, and Iranian earthquakes in 2015, 2010, and 2003, respectively, resulted in approximately 150–200 cases, while the Pakistan earthquake in 2005 yielded more than 600 cases [9]. In 2014, Handicap International identified 370 individuals from the Syrian refugee community with SCI residing in Jordan over a 14-month period (unpublished report).

Increased life expectancy in high income countries has led to higher incidence of SCI due to the associated greater prevalence of spinal degenerative changes with resultant stenosis and ligament ossification, thereby increasing the risk of SCI following trauma such as a fall or hyperextension injury [10]. Higher rates of tetraplegia and paraplegia are observed with both traumatic SCI (TSCI) and non-traumatic SCI (NTSCI), respectively [11]. Globally there is significant disparity and few countries maintain a SCI national registry or national referral pathway. Relatively few specialized SCI rehabilitation centers exist globally and are generally concentrated in high-income countries [12]. The global lack of specialist rehabilitation skills in low-income countries can result in inadequate initial management of SCI cases at the receiving hospital and a rapid development of secondary complications, ultimately contributing to a longer length of hospital stay and potentially poorer functional outcomes or even death [13].

Access to SCI rehabilitation centers varies considerably. In Africa, there are approximately 20 facilities providing SCI rehabilitation, but over half of these are located in South Africa; the majority of African countries remain without an SCI rehabilitation center [14]. In Europe, Australasia, and North America, rehabilitation coverage is considerably higher, and services are configured to support outreach, inpatient rehabilitation, and outpatient follow-up services. Outreach provides early identification and input into management of individuals with SCI until their transfer into specialist services or via transfer of technical knowledge to the treating clinical team. Follow-up post-rehabilitation via a dedicated outpatient service can be delivered via a hybrid of telemedicine and physical appointments, and enables an individual to access support to manage any new health complications that arise or act as a referral into local services.

### 1.1. Highlight: Asia

Access to SCI rehabilitation centers in Asia varies significantly across countries. For instance, India has numerous specialized centers, including the Indian Spinal Injuries Center in Delhi and rehabilitation facilities within large hospital networks [15]. Similarly, Southeast Asian countries such as Thailand, Indonesia, and Malaysia also offer specialized rehabilitation units within major hospitals; however, there are no individual SCI rehabilitation centers [16,17]. Nevertheless, despite the presence of these facilities, many countries in the region still face challenges with access, particularly in rural areas. A study from India highlighted an average 45-day delay between injury and presentation to a spinal unit, largely due to a lack of awareness among healthcare providers about the existence of such centers [15,18]. This illustrates that while rehabilitation centers are available, delay in care remains a critical issue.

In China, where there are around 60,000 new cases of spinal cord injury annually, a SCI network was established in 2004. This Hong Kong-based consortium includes over two dozen centers in mainland China, Hong Kong, and Taiwan, working to test therapies for spinal injuries [19]. Despite challenges with unregulated clinical practices, the network has pushed for better treatment access in the region. A database like Japan’s SCI Database (SCI-J) is vital for improving SCI care. It helps track patient outcomes, identify best practices, and refine treatment protocols. By collecting and analyzing data, healthcare providers can enhance care quality and support research that drives innovation in SCI management. This leads to better patient outcomes and more effective treatments [20].

### 1.2. Highlight: Latin America

More than 79,000 people are affected by SCI each year in Latin America [21]. A review that analyzed data from seven Latin American countries found that road traffic accidents were responsible for 40.8% of SCI cases, with the majority of patients being males in their 30s (76.6%), and the thoracic segments being the most commonly affected (57.9%) [21]. There is incomplete information regarding the management of TSCI during the initial phases, with significant diversity observed among studies; treatment approaches vary widely, and complications contribute to prolonged length of stay, increased expenses, and higher mortality rates [22]. A study in 12 institutions across the region found that most spinal gunshot victims were treated non-surgically, despite the majority having neurological injury or SCI [23].

In Latin America, the availability of SCI rehabilitation services is scarce [24]. A 2022 study concluded that in this region, there was limited professional training and resource availability to assist patients with SCI requiring pulmonary rehabilitation [25]. A study from 2023 that compared SCI patients treated at rehabilitation institutes in Peru and Spain found that the former had greater severity of neurological damage and began their specialized care later [26]. A survey of 318 healthcare professionals from Latin America found that more than 60% reported not feeling prepared to advise patients with SCI around sexuality [27]. Similarly, they rarely carry out interventions that target post-injury mental health [24].

Since 1995, a regional scientific society, the Latin American and Caribbean Spinal Cord Association (ALME), formerly known as the Latin American Society of Paraplegia (SLAP), has worked to enhance the competencies of healthcare professionals to improve the quality of life for SCI patients [28]. This association has four primary aims: (1) to promote the exchange of information among all organizations in the region, (2) to facilitate the dissemination and development of educational resources, (3) to encourage and support efforts aimed at educating, clinical practice, and training healthcare professionals in general, and (4) to promote the development of health observatories, as well as information systems and epidemiological indicators [28]. SCI management is rarely featured in undergraduate curriculums; even many health-related post-graduate programs do not include comprehensive SCI education and training. In Latin America, it is important for medical students to show interest in specializing in this field, to encourage PM&R residents to develop an interest in the respective subspecialty, and to enhance the production of scientific publications on the subject [29]. Some PM&R residency programs in the region include training in SCI management. A study from 2023 that analyzed the 11 PM&R residency programs in Peru found that all of them included the development of competencies in SCI in their curriculums [30]. In contrast, a 2024 study found that out of the 21 PM&R residency training sites in Peru, only 3 offered clinical SCI rotations, leaving many residents unable to meet their program curriculum requirements [31].

## 2. Methods

### 2.1. Design and Development

For this perspective study, we conducted three focus groups between 2023 and 2024 via Zoom (Zoom Video Communications) with authors who are members of organizations belonging to the World Rehabilitation Alliance (WRA) and have clinical experience with SCI patients, to explore rehabilitation professionals’ perspectives on global strategies to strengthen SCI rehabilitation services within health systems. This approach allows us to gain an in-depth understanding of the current initiatives and proposals related to SCI in the context of what concerns the World Health Organization (WHO) and the WRA. After the focus groups, it was agreed to divide the study into four subsegments related to key workstreams of the WRA (Workforce and Education, Health Systems and Policy Research, Primary Care, Emergencies), one on tools and resources for SCI rehabilitation, experiences from national assessments of rehabilitation situations (using WHO products), and a case study. All of the authors participated in the conception and writing of this study, considering their experience, geographical location, and other characteristics.

### 2.2. Context

In 2017, the WHO launched the initiative Rehabilitation 2030 to strengthening rehabilitation in health systems, in alignment with the United Nations Sustainable Development Goal 3, Good Health and Well-Being [32]. Recognizing rehabilitation as a priority health strategy and essential health service, the initiative calls for integration of rehabilitation in all levels of health care, available for all and throughout all ages [33]. Ten areas for action were prioritized, which included developing strong multidisciplinary rehabilitation workforce, building research capacity, and strengthening rehabilitation implementation, including in emergencies [33]. In efforts to accelerate the Rehabilitation 2030 agenda, the WHO established the WRA in 2022 [34]. This global network of over 90 international rehabilitation stakeholders contribute to a global advocacy strategy and further strengthen rehabilitation partnerships inclusive of low-, middle-, and high-income countries [34]. Target advocacy areas of the WRA are workforce, Health Systems and Policy Research (HPSR), primary care, and emergencies [34]. Further, tools and training resources to support national implementation efforts of the 2030 agenda have been disseminated, with additional guidelines and standards developed specifically to address SCI rehabilitation capacity.

## 3. Global Initiatives to Strengthen Rehabilitation in Health Systems

### 3.1. Workforce and Education

SCI rehabilitation requires a multidisciplinary team approach for maximum functional recovery. Workforce development and education for healthcare professionals in SCI management are crucial components in ensuring optimal care for these patients. SCI care emphasizes importance of licensing and certification for practitioners in various disciplines, ensuring that they meet ethical and statutory requirements set by state or national organizations [35]. This is crucial for maintaining high standards of professional conduct and competence. Additionally, educational interventions must prepare the person served and their caregivers to manage healthy routines, maintain safety, and solve issues that commonly occur after SCI [35]. Knowing when and how to access additional assistance and resources in the community and health care system is a critical component of SCI education [35].

Online learning platforms offer accessible and up-to-date resources on SCI management for healthcare professionals seeking to expand their knowledge. Residency programs or rotations in rehabilitation centers or SCI-focused facilities provide practical experience for medical residents. Some of the educational content related to SCI offered by globally recognized institutions is summarized in Table 1.

### 3.2. Health Systems and Policy Research

HPSR can be broadly described as the creation of new knowledge aimed at enhancing the way communities structure themselves to attain health objectives [45]. Health policy research delves into the interactions among diverse stakeholders in policymaking, while health systems research aims to optimize the coverage, quality, efficiency, and equity of health systems (organizations, individuals, and actions dedicated to promote, restore, or maintain health) [45]. HPSR explores contextual factors (both internal and external) and stakeholder concerns, produces necessary data for impactful interventions, and embraces a comprehensive perspective on health and its influencers [46].

HPSR is necessary to improve the rehabilitation of patients with SCI. A study that examined the characteristics of the health systems of 22 countries found that the most common barriers to healthcare access for patients with SCI are the availability and affordability of health services, which predominantly impact countries with limited resources [47]. Since many of the solutions to these barriers that hinder adequate rehabilitation for patients with SCI must be addressed by policymakers, it is essential for researchers to present a comprehensive and flexible plan to influence them, which provides a source of reliable data, considers patients’ advocacy groups and organizations, and emphasizes early interventions [12]. That is to say, based on HPSR.

### 3.3. Primary Care

Evidence indicates that patients with SCI often receive inadequate preventive care and face numerous unmet healthcare needs, with well-documented challenges in accessing optimal primary care stemming from environmental obstacles, such as inaccessible medical facilities and insufficient specialized equipment, as well as a lack of knowledge and academic training regarding SCI issues among primary care providers [48]. SCI patients in primary care should receive at least an annual history and physical examination, along with personalized surveillance tests tailored to their needs; however, the literature does not clearly specify who should coordinate their care, highlighting the need to establish a comprehensive model that adequately addresses both the primary preventive and specialty care required [49].

Several countries have already begun integrating SCI care into their primary care systems. In the United Kingdom, the National Health Service (NHS) emphasizes a community-based model where primary care providers collaborate with specialists to ensure that SCI patients receive ongoing support, especially for bladder and bowel care and pressure ulcer management [50]. The Queensland SCI Service (QSCIS) in Australia is a government initiative designed to improve access to SCI care through primary care. It equips general practitioners with the tools and knowledge to manage common SCI-related issues such as pressure injuries, autonomic dysreflexia, and bowel/bladder care. By providing direct access to specialist teams like the Spinal Outreach Team, QSCIS enables GPs to coordinate more comprehensive care, ensuring timely interventions and support for SCI patients [51]. These examples highlight the importance of integrating SCI management into primary care systems to improve outcomes and the quality of life for those living with SCI.

### 3.4. Emergencies

When a functioning SCI rehabilitation center exists, a 4- or 5-fold increase in cases poses a significant challenge for both staff and infrastructure. In the absence of such a center, there is often no provision for the rehabilitation of SCI cases by emergency medical teams (EMTs), who tend to focus on injuries requiring less long-term skilled care. As observed in Haiti, many people with SCI did not survive, and by the time surviving patients accessed specialized care, many had developed significant, preventable complications [52]. In Pakistan, Rathore described SCI as one of the most neglected conditions, with high rates of secondary complications commonly reported following earthquakes [53,54,55].

Following the Haiti earthquake, international efforts have been made to improve the consistency of access to and the quality of rehabilitation services during emergencies. These efforts have led to better coordination among organizations involved in the response, strengthened cluster coordination mechanisms, categorized EMTs responses into Types 1, 2, and 3, and formalized standards for EMTs, as outlined in the WHO document Minimum Technical Standards for Rehabilitation in Emergencies [56]. Development of global minimum standards for SCI management in emergencies is currently in process following expert stakeholder working groups.

WRA is developing a toolkit for rehabilitation preparedness. This all-hazards toolkit, while not exclusive to SCI, will have direct applications for countries in maintaining essential services during emergencies and managing any potential surge in the demand for rehabilitation, along with possible concurrent damage to infrastructure. Workstreams are currently compiling recommendations for the rehabilitation workforce, maintaining safe rehabilitation infrastructure, and service mapping, with the aim of publication later in 2024.

Globally, there has been a continued demand for SCI rehabilitation services following emergencies. UK-Med is a medical humanitarian charity established in 1995. Drawing on a register of national health service-employed staff, the organization responds to invitations from host ministries of health for technical and material support in responding to global emergencies. As part of the response to the Nepal earthquake in 2015, a team with expertise in SCI was deployed alongside specialist regional teams to support the staff at the Spinal Injuries Rehabilitation Center in Bhaktapur in their surge capacity.

This model enables the rapid upskilling of national staff who may have been recruited in response to the demand for rehabilitation due to a sudden-onset disaster but lack experience in managing SCI, particularly in neurogenic bowel and bladder management, assistive technology prescription and provision, and training in activities of daily living. Deploying a small but highly skilled team representing key multidisciplinary disciplines is a cost-effective option for rapid deployment, and it is anticipated that the impetus provided by the launch of the SCI minimum standards in emergencies will encourage the creation of other SCI-specialized international teams. A further example of this model has been replicated in Ukraine, where a small multidisciplinary team of international experts has provided technical support for strengthening newly developed and existing SCI centers in the country.

ISCoS has a structure of six committees, including a dedicated emergency sub-committee that meets quarterly. This team has been able to provide technical advice and communicate specific requests for expertise or management guidelines in supporting SCI rehabilitation across the ISCoS member network. In 2023, in addition to in-country deployment, educational webinars coordinated by the WHO have been delivered to provide guidance on managing SCI cases in situations where insecurity has prevented physical deployment (Sudan) or where only advice has been requested (Morocco earthquake). These webinars enable just-in-time exchanges, allowing national staff to access a wider international network of expertise for managing complex cases.

### 3.5. Tools and Resources

The Package of Interventions for Rehabilitation (PIR) in SCI is a freely accessible online tool provided by the WHO that comprises evidence-based interventions for people across the care continuum, service delivery platforms, and all world regions, intended for countries to plan, budget, and integrate rehabilitation including assistive technologies and workforce/equipment needs [57]. However, despite having the best available evidence in SCI rehabilitation, Arienti C, et al. noted in 2023 a certain level of uncertainty regarding the effectiveness of some interventions that warrants confirmation through more rigorous studies, so they advised stakeholders to enhance methodological quality in future investigations, taking into account the complexity of this pathology [58].

Essential documents, complementary to the PIR, are SCI rehabilitation clinical practice guidelines from organizations/societies such as AOSpine, Canadian SCI Practice (Can-SCIP), and Paralyzed Veterans of America (PVA), among others [59,60,61]. Another valuable resource for academics and the lay public can be found on the websites of the ASIA, the ISCoS, or the SCI Research Evidence (SCIRE) Professional, which encompass worksheets, articles, outcome measures, podcasts, videos, certified online courses, interest groups, research and grant opportunities, and much more, available in various languages including English, French, Spanish, Portuguese, and Greek [62,63,64].

### 3.6. National Assessment

In many countries, access to rehabilitation services is constrained by a lack of skilled personnel and limited availability of assistive technology. Furthermore, the sector often suffers from weak leadership, insufficient government funding, and, in some cases, a lack of clarity about the necessary resources. To address these challenges, the WHO released the *Rehabilitation in Health Systems: Guide for Action* in 2019. This comprehensive, four-step tool is designed to help nations assess their needs and formulate a strategic plan to improve the accessibility, quality, and effectiveness of rehabilitation services [65].

The WHO Systematic Assessment of Rehabilitation Situation (STARS) is part of this tool, and as of September 2021, it had guided 21 situation assessments in countries such as Jordan, Myanmar, Sri Lanka, Solomon Islands, Laos, Haiti, and Guyana, with the involvement of government representatives, consultants, staff from WHO country or regional offices, and experts [66]. Peru, another low- and middle-income country (LMIC), under the leadership of the PM&R Department of Hospital Nacional Edgardo Rebagliati Martins (HNERM), a member of the WRA, has begun using this tool to prepare a rehabilitation workforce evaluation report, similar to the one conducted in Poland [67], as a component of a program aimed at enhancing collaborative work, patient care, and academic pursuits [68].

## 4. Case Study: Peru

Here, presented is the experience of Peru, a middle-income country from Latin America, conducting a structured national rehabilitation assessment using tools developed by the WHO, in response to their Rehabilitation 2030 initiative.

### 4.1. Introduction

In Latin America and the Caribbean, a region comprising LMICs, an increasing prevalence of non-communicable chronic diseases is projected, including conditions and disorders which often lead to disability and require rehabilitation [69]. The strengthening of rehabilitation services has become a pressing priority in LMICs, where up to half of the population lacks access to necessary care [66,69]. The WHO developed the STARS in 2019, the first health system assessment tool designed to aid governments in gaining a better understanding of the state of rehabilitation within their respective countries, to identify strengths, gaps, and priorities [66,70].

### 4.2. Local Situation, Tools, and Resources

Peru has been represented in the WRA since May 2023 through the PM&R Department of HNERM, one of the most important national referral centers of the Social Health Insurance of Peru (EsSalud), responsible for leading and advocating for PM&R in the region [68]. Subsequently, in August 2023, the PM&R Department of HNERM established its own WRA Program, an internal enhancement framework inspired by the Alliance’s structure, with one of its goals being to develop a national rehabilitation assessment report in partnership with the Pan American Health Organization (PAHO) [68].

In February 2024, two important documents related to disability and rehabilitation were published in Peru: the document “Characterization of Living Conditions of the Population with Disabilities, 2022” from the National Institute of Statistics and Informatics of Peru (INEI), which highlights the main aspects of the living conditions of people with disabilities [71], and the 2024–2025 Work Plan developed by the PAHO/WHO Representation Office in Peru [72]. This plan outlines the products and services of technical cooperation with the country, with a commitment to immediate, intermediate, and impact results, including the development of plans, programs, and strategies on rehabilitation and assistive technology, particularly in emergency contexts.

### 4.3. Objectives

The objective of the PM&R Department of HNERM was to initiate a national rehabilitation assessment report, which had never been conducted in Peru. Physiatrists from the Department, representing the WRA, reached out to PAHO during the last quarter of 2023, achieving an informal agreement to begin collaboration. In July 2024, after several meetings with various stakeholders, the Department sent a letter to PAHO to formalize the technical cooperation, involving the Ministry of Health of Peru (MINSA), as the country’s leading health authority.

### 4.4. Results

In August 2024, technical cooperation was formalized under the leadership of MINSA, with guidance from PAHO and the WHO. It was agreed to create a working group with representatives from the main health institutions in Peru, holding monthly virtual meetings to advance a common work plan. In the initial meeting, it was decided to use the STARS tool for this purpose. To date, the working group has completed STARS Step 1 (prepare for situation assessment) and is currently conducting STARS Step 2 (collect data and information) using the Template for Rehabilitation Information Collection (TRIC) tool [65].

## 5. Limitations and Future Research Directions

This perspective study has several limitations. Due to the wide range of professionals involved in SCI rehabilitation, it was not feasible to include all relevant professions, which may affect the external validity of the findings. A broader and more diverse selection of participants could have revealed additional insights. However, this limitation is likely to have had a positive rather than restrictive impact on the quality of the research. Additionally, it is important to note that the authors also served as data collectors, analysts, and reporters, which may introduce potential biases.

Rehabilitation has garnered growing global attention, and the demand for rehabilitation services is expected to rise in the future. This study highlights the current needs in SCI rehabilitation, particularly in terms of services, education, and research. However, significant knowledge gaps remain. Future research should focus on understanding the evolving needs of key stakeholders, including healthcare professionals, patients, and other relevant actors, as well as exploring effective strategies to address and meet these needs.

## 6. Conclusions

The WRA’s commitment to SCI rehabilitation as an essential health service aligns with global health initiatives such as the WHO’s Rehabilitation 2030. Our analysis reveals the pressing need for SCI rehabilitation services worldwide, particularly in LMICs where resources, specialized centers, and trained personnel remain scarce. Through strategic collaboration and resource-sharing, international organizations like the WHO and WRA play a pivotal role in bridging these gaps by providing standardized tools, competency-based frameworks, and technical guidelines aimed at enhancing national rehabilitation capacities. The findings in this perspective study underline the impact of SCI on individuals’ physical, mental, and socioeconomic well-being and the far-reaching implications for families and communities. A robust, multidisciplinary approach to SCI rehabilitation not only mitigates secondary complications, but also significantly enhances quality of life and societal integration for those affected. Through targeted workforce development, the establishment of national registries, and the integration of SCI care into primary and emergency services, we can achieve a more equitable and effective global health landscape. The ongoing development and implementation of WHO-supported tools, such as the PIR and the STARS assessment framework, offer countries the means to identify local needs, plan services, and align with global best practices. In collaboration with WRA, health systems worldwide are better equipped to mobilize resources and adapt services to meet rising demand, thus moving closer to the goal of comprehensive, accessible rehabilitation for all.

## Figures and Tables

**Table 1 healthcare-12-02313-t001:** Educational content related to SCI offered by globally recognized institutions.

**Institution**	**Educational Content**
University of Alberta	Offers a 6-week online SCI rehabilitation course with 35 educational hours, covering key impairments and management strategies through patient videos [36].
Spinal Injuries Association (UK)	Provides short online courses on SCI topics, including communication, PTSD, and psychological and emotional health [37].
International Spinal Cord Society (ISCoS)	Developed a Knowledge and Skills Framework for SCI, complementing PM&R training. The ISCoS Toolkit for Strengthening SCI Services is being piloted to support government and rehabilitation actors in evaluating or initiating SCI services [38].
Spinalistips by Spinalis Foundation	Offers a platform for people with SCI to share practical advice on assistive devices and adaptations, promoting self-reliance [39].
Shepherd Center (USA)	Provides resources for SCI patients on functional recovery, sensory issues, and bladder/bowel management [40].
Academy of Neurologic Physical Therapy (ANPT)	Through its SCI Special Interest Group, ANPT provides manuals, research opportunities, and funding resources for healthcare providers, patients, and families [41].
American Spinal Injury Association (ASIA)	Offers online courses on SCI, including self-paced multimedia modules with assessments and certificates covering autonomic function, skincare, spasticity, urology, and pediatric topics [42].
Indian Spinal Injuries Center and Indo-American Spine Alliance (IASA)	Offers a 6-month fellowship for orthopedic and neurosurgeons with a focus on spine surgery, including opportunities to train at international institutions approved by IASA [43].
Academy of SCI Professionals	Offers a specialized SCI fellowship with certification for practitioners in relevant specialties, including anesthesiology, PM&R, emergency medicine, neurology, surgery, pediatrics, and urology [44].

## Data Availability

No new data were created or analyzed in this study.

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
