# Peer review of "Perspective of the World Rehabilitation Alliance: Global Strategies to Strengthen Spinal Cord Injury Rehabilitation Services in Health Systems"

_healthcare, 2024, doi:10.3390/healthcare12222313_

Round 1

Reviewer 1 Report

Comments and Suggestions for Authors

Dear Authors,

Thank you for submitting your perspective paper on global strategies to strengthen spinal cord injury rehabilitation services. Your work provides a comprehensive and timely overview of this important topic. The paper's structure is logical and effective, moving from background information to specific global efforts and concluding with a valuable case study from Peru.

Your global perspective, incorporating information from various countries and regions, is a key strength of this work. The linkage to broader health initiatives, such as the WHO's Rehabilitation 2030 agenda, effectively demonstrates the relevance of SCI rehabilitation in the larger context of global health. The inclusion of specific tools, resources, and a case study adds practical value to the paper.

To further strengthen your excellent work, consider adding a brief methods section explaining how information was gathered and synthesized. Additionally, a short discussion of limitations and future research directions would enhance the paper's impact.

Overall, your perspective piece makes a valuable contribution to the field and will be of great interest to healthcare professionals and policymakers worldwide.

Author Response

Dear Reviewer 1,

Comment 1: "To further strengthen your excellent work, consider adding a brief methods section explaining how information was gathered and synthesized."

Response: We appreciate the comment. We have added a brief Methods section where we explain the conception, design, and development of the article.

Comment 2: "Additionally, a short discussion of limitations and future research directions would enhance the paper's impact."

Response: We appreciate the comment. We have added a Limitations and Future Research Directions section.

Reviewer 2 Report

Comments and Suggestions for Authors

After completion of the assessment work on this manuscript entitled on 'Perspective of the World Rehabilitation Alliance: Global Strategies to Strengthen Spinal Cord Injury Rehabilitation Services in Health Systems', in current form it is unsuitable to recommend for publication due to minor query with respect to methodology of the study and its scientific discussion in results of case study as well as poor conclusion.

1. Improve abstract with respect to methodology and conclusion of the study.

2. The case studies are required to improve with respect to their objectives and results.

3. The meaningful conclusion is required to add in conclusion section rather than simple sentences of the manuscript.

Hence, minor revision is essential.

Comments on the Quality of English Language

English language is good for ease of understanding. No need to improve in language.

Author Response

Dear Reviewer 2,

Comment 1: "Improve abstract with respect to methodology and conclusion of the study."

Response: We appreciate the comment. We have improved the Abstract by adding the relevant information about the Methods and the new Conclusions.

Comment 2: "The case studies are required to improve with respect to their objectives and results."

Response: We appreciate the comment. We added subsections for Objectives and Results to the Case Study.

Comment 3: "The meaningful conclusion is required to add in conclusion section rather than simple sentences of the manuscript."

Response: We appreciate the comment. We revised all the Conclusions to make them more coherent with what we aim to convey through the study.

Reviewer 3 Report

Comments and Suggestions for Authors

This article is an overview of the WHO Rehabilitation 2030 agenda adapted to the management of spinal cord injuries worldwide. After an epidemiological presentation of spinal cord injuries, focusing on two continents, the authors describe some real-world educational content and how to optimize management at each stage of spinal cord injuries.

I just have a few remarks :

- line 71, it seems that a verb is lacking "or conflict may ? a higher incidence"

- line 76, there is comma between "period, (unpublished report)"

- line 92, it seems that a comma is lacking "Australasia and North America, rehabilitation coverage"

- paragraph 2.1 Workforce and education. Presentation of educational content (University of Alberta, ISCoS, Spinalistips initiative, Shepherd center, ANPT, Indian Spinal Injury Centre, Academy of SCI Professionals) could be more readable in a table format. 

- line 294, the acronym WRA is not defined

Author Response

Dear Reviewer 3,

Comment 1: "line 71, it seems that a verb is lacking "or conflict may ? a higher incidence"

Response: We appreciate the comment. The error has been corrected in the manuscript.

Comment 2: "line 76, there is comma between "period, (unpublished report)"

Response: We appreciate the comment. The error has been corrected in the manuscript.

Comment 3: "line 92, it seems that a comma is lacking "Australasia and North America, rehabilitation coverage"

Response: We appreciate the comment. The error has been corrected in the manuscript.

Comment 4: "paragraph 2.1 Workforce and education. Presentation of educational content (University of Alberta, ISCoS, Spinalistips initiative, Shepherd center, ANPT, Indian Spinal Injury Centre, Academy of SCI Professionals) could be more readable in a table format."

Response: We appreciate the comment. We present the content in a table to make it more readable (Table 1).

Comment 5: "line 294, the acronym WRA is not defined"

Response: We appreciate the comment. We have added the definition of WRA in the Methods section.